# Cohort profile: Study on Zika virus infection in Brazil (ZIKABRA study)

Guilherme Amaral Calvet[1]☯*, Edna Oliveira Kara[2]☯, Sihem Landoulsi[2], Ndema Habib[2], Camila Helena Aguiar Bôtto-Menezes[3,4], Rafael Freitas de Oliveira Franca[5], Armando Menezes Neto[5], Marcia da Costa Castilho[3], Tatiana Jorge Fernandes[1], Gerson Fernando Pereira[6], Silvana Pereira Giozza[6], Ximena Pamela Díaz Bermúdez[7], Kayvon Modjarrad[8], Noemia Lima[8], Patrícia Brasil[1], Marcus Vinicius Guimarães de Lacerda[3,9], Ana Maria Bispo de Filippis[10], Nathalie Jeanne Nicole Brutet[2], on behalf of ZIKABRA Study Team¶

1 Acute Febrile Illnesses Laboratory, Evandro Chagas National Institute of Infectious Diseases, Oswaldo Cruz Foundation, Rio de Janeiro, Brazil, 2 Department of Sexual and Reproductive Health and Research, World Health Organization, Geneva, Switzerland, 3 Department of Malaria, Tropical Medicine Foundation Doctor Heitor Vieira Dourado (FMT-HVD), Manaus, Amazonas, Brazil, 4 School of Health Sciences, Amazonas State University (UEA), Manaus, Amazonas, Brazil, 5 Department of Virology and Experimental Therapy, Institute Aggeu Magalhães, Oswaldo Cruz Foundation, Recife, Pernambuco, Brazil, 6 Department of Chronic Condition Diseases and Sexually Transmitted Infections, Health Surveillance Secretariat, Ministry of Health, Brasilia, Brazil, 7 Department of Public Health, University of Brasilia, Brasília, Brazil, 8 Emerging Infectious Diseases Branch, Walter Reed Army Institute of Research, Silver Spring, MD, United States of America, 9 Instituto Leônidas & Maria Deane, Oswaldo Cruz Foundation, Manaus, Amazonas, Brazil, 10 Flavivirus Laboratory, Oswaldo Cruz Institute, Oswaldo Cruz Foundation, Rio de Janeiro, Brazil

☯ These authors contributed equally to this work.
¶ Membership of the ZIKABRA Study Team is listed in the Acknowledgments.
* guilherme.calvet@ini.fiocruz.br

**Data Availability Statement:** Data underlying the study cannot be made publicly available due to ethical concerns, as data contain several personally identifiable information. Data are available from Oswaldo Cruz Foundation for researchers who

## Abstract

Zika virus (ZIKV) has been detected in blood, urine, semen, cerebral spinal fluid, saliva, amniotic fluid, and breast milk. In most ZIKV infected individuals, the virus is detected in the blood to one week after the onset of symptoms and has been found to persist longer in urine and semen. To better understand virus dynamics, a prospective cohort study was conducted in Brazil to assess the presence and duration of ZIKV and related markers (viral RNA, antibodies, T cell response, and innate immunity) in blood, semen, saliva, urine, vaginal secretions/menstrual blood, rectal swab and sweat. The objective of the current manuscript is to describe the cohort, including an overview of the collected data and a description of the baseline characteristics of the participants. Men and women ≥ 18 years with acute illness and their symptomatic and asymptomatic household contacts with positive reverse transcriptase-polymerase chain reaction test for ZIKV in blood and/or urine were included. All participants were followed up for 12 months. From July 2017 to June 2019, a total of 786 participants (284 men, 502 women) were screened. Of these, 260 (33.1%) were enrolled in the study; index cases: 64 men (24.6%), 162 (62.3%) women; household contacts: 12 men (4.6%), 22 (8.5%) women. There was a statistically significant difference in age and sex between enrolled and not enrolled participants (p<0.005). Baseline sociodemographic and medical data were collected at enrollment from all participants. The median and interquartile range (IQR) age was 35 (IQR; 25.3, 43) for men and 36.5 years (IQR; 28, 47) for women.

meet the criteria for access to confidential data. Contact information: Institutional Ethics and Research Committee of the Evandro Chagas National Institute of Infectious Diseases, email: cep@ini.fiocruz.br or Guilherme Amaral Calvet; email: guilherme.calvet@ini.fiocruz.br.

**Funding:** GAC, AMBF, KM, MVGL, NJNB. The research leading to these results received funding from the Wellcome Trust: Grant Number 206522/Z/17/Z, World Health Organization: Reference TSA1-2017/720873-0 and TSA2-2017/731359-0, Brazilian Ministry of Health: Convênio 837059/2016, Processo 25000162039201616, National Institute Of Allergy And Infectious Diseases of the National Institutes of Health: Award Number R21AI139777 and the Henry M. Jackson Foundation for the Advancement of Military Medicine: Prime Award No W81XWH-18-2-0040. The funders had no role in study design, data collection, and analysis, decision to publish, or preparation of the manuscript.

**Competing interests:** The authors have declared that no competing interests exist.

Following rash, which was one of the inclusion criteria for index cases, the most reported symptoms in the enrollment visit since the onset of the disease were fever, itching, arthralgia with or without edema, non-purulent conjunctivitis, headache, and myalgia. Ten hospitalizations were reported by eight patients (two patients were hospitalized twice) during follow up, after a median of 108 days following symptom onset (range 7 to 266 days) and with a median of 1.5 days (range 1 to 20 days) of hospital stay. A total of 4,137 visits were performed, 223 (85.8%) participants have attended all visits and 37 (14.2%) patients were discontinued.

## Introduction

The Zika virus (ZIKV) has been found in body fluids such as blood, urine, semen, brain and spinal fluids, saliva, rectal swab, vaginal secretions, amniotic fluid, and breast milk [1–8]. Studies have found that the ZIKV RNA can persist longer in urine and semen when compared to its presence in blood [5, 7, 9]. This is the first virus known to be transmitted to humans through the bite of an infected mosquito and sexual intercourse with an infected person [10, 11].

Since 2016, the World Health Organization (WHO) and the Brazil Ministry of Health (MoH) are coordinating the ZIKABRA study, a research cohort study in Brazil, set up to address questions such as how long does the virus remains in the body? How long infectivity can take place? Could the virus remain inactive in a person and reappear at a later stage? What are the potential relations between host genetics, immune response, and environmental factors as well as co-infections such as dengue (DENV), chikungunya (CHIKV), human immunodeficiency virus (HIV), syphilis, Hepatitis B (HBV), hepatitis C (HCV), and the duration of the persistence of the virus in body fluids? The responses will help to refine recommendations on how best to prevent ZIKV virus infection.

The ZIKABRA is a prospective cohort study which aims to assess the presence and duration of ZIKV and related markers (ZIKV specific RNA, antibodies, T cell response, and innate immunity) in blood, semen, saliva, urine, vaginal secretions/menstrual blood, rectal swab and sweat. The study has been conducted in collaboration with the Brazil Ministry of Health, Oswaldo Cruz Foundation (Fiocruz), Tropical Medicine Foundation Doctor Heitor Vieira Dourado (FMT-HVD), and the Walter Reed Army Institute of Research (WRAIR). The study was also supported by grants from the Wellcome Trust and the National Institute of Allergy and Infectious Diseases of the National Institutes of Health.

This paper aims to provide a comprehensive description of the ZIKABRA cohort, including an overview of the collected data and a description of the baseline characteristics of the participants.

## Material and methods

### Index screening inclusion and exclusion criteria

Index case screening inclusion criteria: Men and women $\geq$ 18 years with acute illness presenting rash; having given consent to blood and urine collection for real time revere transcription polymerase chain reaction (rRT-PCR) testing and to be enrolled if the test result in either sample return positive, including attendance to the study clinic for follow-up visits over 12 months comprising 17 visits, and provide body fluid according to the testing schedule; no intention to

move to a place that would not make possible attendance to the study clinic to continue with the study procedures as per protocol.

Index case screening exclusion criteria: Individuals under 18 years of age; pregnancy; presenting a condition that would not allow reliable informed consent (e.g. alcohol abuse and substance misuse) or lacking mental capacity to consent participation.

Index case enrolment inclusion criteria: Presenting rRT-PCR test positive for ZIKV in blood and/or urine specimens; having given consent for all body fluid collection, as appropriate, according to the testing schedule.

Index case enrolment exclusion criteria: Presenting rRT-PCR negative test results for ZIKV in blood and urine specimens collected during screening.

## Household contacts screening and enrollment criteria

Household contact screening inclusion criteria: individual aged 18 years and above; having given consent for rRT-PCR testing in blood and urine and subsequent enrollment if the tests return a positive result, including coming back to all follow-up visits and provide body fluid according to the testing schedule; no intention to move to a place that would not make possible attendance to the study clinic to continue with the study procedures as per protocol.

Household contact screening exclusion criteria—As per index case.

Household contact enrolment inclusion and exclusion criteria- As per index case.

## ZIKV detection

ZIKV detection was performed by rRT-PCR employing a commercial kit namely ZDC, from the Instituto de Tecnologia em Imunobiológicos Biomanguinhos. The kit was approved by the Agência Nacional de Vigilância Sanitária/ANVISA (registry #80142170032). A test was deemed positive when a sample returned within 38 amplification cycles.

The detailed study protocol has been published elsewhere [12]. Since then, some changes were performed in the study case report forms (CRFs). The final versions of the CRFs can be found in the supplement information.

## Screening and recruitment in the cohort

Patients with ZIKV infection were screened and enrolled in two cities, Recife and Manaus, representing two regions of Brazil (Northeast and North, respectively). The study sites were selected based on the presence of high population density, high circulation of ZIKV, strong community health network and laboratory facilities able to perform viral culture, ZIKV antigen assays, rRT-PCR, IgM/IgG, neutralizing antibodies test (specific for ZIKV, DENV, and CHIKV) and genetic sequencing. However, by the time of the study onset, the number of ZIKV cases dropped dramatically in Recife. As a result, only five participants were recruited in that site between 21 July 2017 and 13 August 2018.

The identification of symptomatic subjects took place in the collaborating clinics: Fiocruz ambulatory (Recife), Tropical Medicine Foundation Doctor Heitor Vieira Dourado (FMT-HVD) ambulatory (Manaus), primary health centers (Family Clinics) and 24-hour emergency units located in the areas of the study intervention. At each of these screening sites, individuals meeting the study inclusion criteria were referred to the study sites for consideration for enrolment.

**Case report forms, procedures, and data collection.** Eleven CRFs were elaborated to collect the study information for each participant. Table 1 shows a summary of the forms and related study visits.

**Table 1. Summary of questionnaires and data collection for ZIKABRA study.**

| Form Name | Visit Number | | | | | | | | | | | | | | | | | |
|---|---|---|---|---|---|---|---|---|---|---|---|---|---|---|---|---|---|---|
| | V0 | V1 | V2 | V3 | V4 | V5 | V6 | V7 | V8 | V9 | V10 | V11 | V12 | V13 | V14 | V15 | V16 | V17 |
| PRE | X | | | | | | | | | | | | | | | | | |
| TRI | X | | | | | | | | | | | | | | | | | |
| IDE | X | | | | | | | | | | | | | | | | | |
| REC | | X | | | | | | | | | | | | | | | | |
| AMO | | X | X | X | X | X | X | X | X | X | X | X | X | X | X | X | X | X |
| RES | X | X | X | X | X | X | X | X | X | X | X | X | X | X | X | X | X | X |
| RAP | | X | | | | | | | | | | X | | | | | | X |
| CLA | | X | | | | X | | | | | | | | | | | | X |
| SEG | | X | X | X | X | X | X | X | X | X | X | X | X | X | X | X | X | X |
| CLB | | | X | X | X | | X | X | X | X | X | X | X | X | X | X | X | |
| FIM | | | | | | | | | | | | | | | | | | X |

PRE (Pre-Screening): Type of potential participant (index/contact), relationship with the index case, and eligibility for screening.

TRI (Screening): Symptoms of ZIKV infection.

IDE (Identification): Sensitive participant information and linkage with the national database.

REC (Enrollment): Sociodemographics data, biological sex, race/ethnicity, formal education, marital status, and vital signs.

AMO (Sample collection): Body fluids collection (blood, urine, saliva, sweat, rectal swab, semen, vaginal, and breastmilk, if available.

RES (Results): Body fluids results.

RAP (Rapid tests/Serology): HIV, syphilis, pregnancy, HBV, and HCV tests.

CLA (Clinical questionnaire A): Comorbidities, history of arboviruses infection (DENV, ZIKV, CHIKV and YF), history of immunization, signs, and symptoms, results for exams requested or referral to specialists, physical examination, other clinical or laboratory diagnosis, exams results, and referral to specialists.

SEG (Follow-up): New cases in the household, any hospitalization, health and wellbeing, signs of reactivation, reinfection, or complications of ZIKV infection, and vital signs.

CLB (Clinical questionnaire B): Intermediary visits as needed, results for exams requested or referral to specialists, symptoms of ZIKV reinfection/reactivation or complication, new symptoms, physical examination, other clinical or laboratory diagnosis, exams results, and referral to specialists.

FIM (End of the study): End of the study participation or reasons for premature participant discontinuation.

**Pre-screening and screening visit.** Two informed consent forms (ICF) were used in the study, for the screening and enrollment phases. After ensuring the message addressed to participants was clearly understood and clarification to their questions and information details had been updated, an initial screening informed consent was signed. During the pre-screening visit (S1 File) the type of potential participant (index/household contact), relationship with the index case, symptoms of ZIKV infection (S2 File), and eligibility were assessed by the study nurse who checked whether they met the inclusion criteria and did not meet any of the exclusion criteria. Sensitive participant information and linkage with the national database were recorded (S3 File). After that, blood and urine were collected for ZIKV testing.

The potential participants were also advised that if their test yielded a positive result, and if they agreed, their household contacts would be invited to participate in the study following the same procedures.

**Enrollment visit.** The screening test results were delivered to the participant by the study nurse. Individuals whose ZIKV rRT-PCR tests returned negative in blood and urine, were referred to the clinic where they were seen in the first place to receive routine clinical standard of care. Individuals with rRT-PCR tests positive in blood or urine were classified as index cases and were invited by the study nurse to be part of the one-year follow-up. At this stage, they were asked to sign a full study informed consent (recruitment), and an enrollment questionnaire (S4 File) was applied to collect a more detailed sociodemographic data, biological sex,

race/ethnicity, formal education, marital status, and vital signs. Specimen collection of all body fluids was then performed. Participants provided blood, urine, saliva, sweat, rectal swab, semen specimens (male participants), or vaginal and menstrual blood specimens (female participants) and breast milk specimens (if lactating). Detailed information on specimen collection, including visit date, visit number, type of visit (if scheduled or non-scheduled), and reason, if not collected, were noted on the sample collection form (S5 File) and the test results recorded in the results form (S6 File). Test results for HIV, syphilis, pregnancy, HBV, and HCV were recorded in the rapid tests/serology form (S7 File) at enrollment, six and 12-months post-enrollment.

All study participants were evaluated by a study physician that collected a detailed baseline clinical information (S8 File) that included comorbidities, history of arboviruses infection (DENV, ZIKV, CHIKV, and yellow fever (YF)), history of immunization, clinical manifestations, results for exams requested or referred to specialists, physical examination, other clinical or laboratory diagnosis, exams results, and referral to specialists. The clinical questionnaire form (S8 File) was completed at the enrolment (visit 1), twenty days (visit 5), and 360 days (visit 17) after the enrollment visit.

Participants were followed for 12 months to evaluate the persistence of ZIKV at 2, 4, 10, 20, 30, 60, 90, 120, 150, 180, 210, 240, 270, 300, 330, and 360 days following the recruitment visit and specimen collection. Participants who could not be contacted after three attempts (phone contact) or did not present to the study clinic for two consecutive visits were considered as lost to follow-up. At each follow-up visit information on new cases in the household, hospitalization, health and wellbeing, signs of reactivation, reinfection, or complications of ZIKV infection, and vital signs were noted in the follow-up questionnaire (S9 File). The sample collection form (S5 File) and results form (S6 File) were also completed in each study visit.

If the participant presented complications during any follow up visit that required medical evaluation, a new form was completed (S10 File). This questionnaire captured results for exams requested or referral to specialists, symptoms of ZIKV reinfection/reactivation or complication, new symptoms, new signs in physical examination, other clinical or laboratory diagnosis and exam results. At the end of the study participation or in case of premature discontinuation, the end of the study form (S11 File) was filled.

Household contact who consented and had detectable ZIKV rRT-PCR in blood and or urine collected in the screening visit, followed the same procedures described above.

## Samples storage

A biorepository, linked to the ZIKABRA protocol, was created in the participant laboratories aiming at the possibility of use in future investigations. According to the Brazil ethical regulation (regulatory documents and resolutions) resolutions, the term of validity of this type of biorepository can be authorized for up to 10 years, being possible renewals authorized by the relevant ethics committees (local review board/national review board) through an examination of justification and report presented by the researcher. The study will follow the norms contained in the Resolution CNS 441/2011 and Ordinance MS 2201.2011 of the Ministry of Health.

## Statistical analyses

The study data were collected and managed using REDCap (Research Electronic Data Capture) tools hosted at Evandro Chagas National Institute of Infectious Diseases, Fiocruz, Rio de Janeiro, Brazil [13, 14].

Statistical analyses of the study cohort profile were performed using IBM SPSS Statistics 22.0. The sociodemographic and clinical variables were determined using frequencies and

proportions for categorical variables and medians and either ranges or Interquartile Ranges (IQRs) for continuous variables. Chi-square test or Fisher's exact test, for categorical variables, and Wilcoxon-Mann-Whitney test for continuous variables, were conducted to assess differences in sex and age between individuals enrolled and not enrolled in the study and to compare characteristics of participants who were discontinued with those who remained in the study. Two-sided tests with 5% significance levels and 95% confidence intervals were used for all parameters.

## Ethical approvals

The study protocol and procedures were reviewed and approved by the WHO Ethics Review Committee (WHO ERC), Protocol ID: ERC.0002786; the Brazilian National Research Ethics Commission (CONEP)(CAAE: 62518016.6.1001.0008); the Institutional Ethics and Research Committee of the Evandro Chagas National Institute of Infectious Diseases, Fiocruz, Rio de Janeiro (CAAE: 62518016.6.2002.5262), the Ethics and Research Committee of the Rio de Janeiro's Municipal Secretary of Health (CAAE: 2518016.6.3001.5279); the Institutional Ethics and Research Committee of the Aggeu Magalhães Research Center, Fiocruz, Recife (CAAE: 62518016.6.2001.5190) and the Institutional Ethics and Research Committee of the Tropical Medicine Foundation, Manaus, Amazonas (CAAE: 62518016.6.2003.0005). In addition, a memorandum of understanding between the Brazil MoH, WHO, Fiocruz, and WRAIR was signed.

## Results

Between July 2017 and June 2019, 786 patients were assessed for eligibility (Fig 1). Of these, 260 (76 men, 184 women) participants were enrolled in the study, 255 in Manaus, and five in Recife. There was a statistically significant difference in age and sex among enrolled and not enrolled participants (Table 2).

A total of 4,137 study visits were performed (4,097 scheduled and 40 unscheduled visits). Among the 260 enrolled patients, 226 (86.9%) were identified as index cases and 34 (13.1%) were household contacts.

## Demographic and clinical characteristics

The demographic and clinical characteristics of the study cohort are shown in Table 3. The cohort consists mainly of women (70.8%), single (46.7%), with a median age of 36.5 years (IQR; 28, 47), and with mixed-race/ethnicity (75.5%). Most individuals had high school/technical college and university/post-graduation levels of education (82.6%). Prevalence of baseline overweight and obesity were remarkably high in the study population (72.8%). No study participants were diagnosed with yellow fever in the past or malaria within 30 days prior to enrollment.

Seven men (9.2%) and 38 women (20.7%) were regularly taking medications for more than 30 days. Male patients were taking medications mainly for diabetes, hypertension, coronary heart disease, and dyslipidemia, while women were being regularly medicated for hypertension, contraception, menopause, thyroid disorders, gastritis, and depressive symptoms.

Regarding yellow fever vaccine history, 50 men (65.8%) and 131 women (71.2%) received at least one dose, 21 men (27.6%) and 38 women (20.7%) did not know whether they were vaccinated, and only five men (6.6%) and 15 women (8.2%) were not immunized.

During adulthood, one man had immunization history against hepatitis A (HAV) (1/76–1.3%), seven against HBV (7/76–9.2%), seven against measles (7/76–9.2%), five against rubella (5/76–6.6%), 16 against tetanus/diphtheria (16/76–21.1%), and nine against seasonal influenza

**Screening**

Assessed for eligibility (n = 786)

Index case = 678 (Male = 230, Female = 448)

Household contact = 108 (Male=54, Female =54)

Excluded (n= 526)

Age <18 years (n=19)
Does not live in an area that allows
    for frequent study visits (n= 61)
Unable to provide consent (n= 4)
Pregnancy (n= 15)
Positive rRT-PCR test, but declined
    to participate (n=31)
Negative rRT-PCR test (blood and
    urine (n= 396)

**Enrollment**

Enrollment (n = 260)

Index case = 226 (Male = 64, Female =162)

Household contact = 34 (Male =12, Female = 22)

Reasons for discontinuation (n= 37)

Moved to an area far from the study
center= 7

Declined to continue participation = 5

Became pregnant = 6

Man unable to provide semen sample
for two consecutive visits = 2

Lost to follow-up, unknown reason =17

**Follow-Up**

223 patients with completed study visits

**Fig 1. Flow diagram of ZIKABRA study.**

(9/76–11.8%). No man was immunized with the human papillomavirus (HPV) vaccine. In contrast, 97 women had immunization history of HBV (40/184–21.7%), measles (24/184–13.0%), rubella (25/184–13.6%), tetanus/diphtheria (81/184–44.0%), and seasonal influenza (48/184–26.1%). Only two women were immunized against HPV. No woman was immunized against HAV and just only one man received a dengue vaccine.

Following rash, which was one of the inclusion criteria for index cases, the most reported symptoms since the onset of the disease were fever, itching, arthralgia with or without edema, non-purulent conjunctivitis, headache, and myalgia. Hemorrhagic symptoms (epistaxis, gingival, and metrorrhagia) were reported in six women. Two asymptomatic participants were included as contacts (one man and one women) (Table 4).

Study participant's examination findings at enrollment included rash, non-purulent conjunctivitis, periarticular edema, and lymphadenomegaly, usually observed in patients with ZIKV infection (Table 5).

Ten hospitalizations were reported by eight patients (two patients were hospitalized twice) during follow up, after a median of 108 days (range 7 to 266 days) following symptom onset and with a median of 1.5 days (range 1 to 20 days) of hospital stay. Reasons for hospitalizations were surgery for a teratoma removal, surgery for the extraction of a cyst in the left ovary, immobilization for fracture of the left clavicle and fracture of three fingers of the left hand, clinical and support treatment for cholelithiasis, generalized anxiety disorder, myopericarditis, severe headache, and unspecified otalgia. No deaths were associated with ZIKV infection in this cohort.

## Attrition during follow-up

Thirty-seven participants were discontinued (14.2%) (Fig 1), of which 17 were lost to follow-up for unknown reasons, seven moved to an area far from the study center, six women became pregnant, five declined to continue further participation, and only two men were unable to provide a semen sample for two consecutive visits. This late discontinuation criterion was withdrawal after a protocol amendment. There was a statistically significant difference between the sex (p = 0.043) and the age (p = 0.012) of discontinued participants with participants who remained in the study (Table 6).

## Strengths and limitations

### Strengths

Having enrolled 260 men and women with confirmed ZIKV infection and with one year of follow-up after the acute infection, the ZIKABRA study is the longest cohort study to investigate the presence and persistence of ZIKV in several body fluids.

**Table 2. Age and sex of 786 individuals assessed for eligibility in the ZIKABRA study.**

| Characteristics | Enrolled | Not Enrolled | P value |
|---|---|---|---|
|  | (*n* = 260) | (*n* = 526) |  |
| Age, years, median (IQR) | 36 (27–46) | 31 (23.8–41) | <0.001 |
| Sex, *n* (%) |  |  |  |
| Men | 76 (29.2) | 208 (39.5) | <0.005 |
| Women | 184 (70.8) | 318 (60.5) | <0.005 |

**Table 3. Baseline characteristics of 260 study participants of the ZIKABRA study, by sex.**

| Characteristics | Male | Female |
|---|---|---|
|  | (*n* = 76) | (*n* = 184) |
| **Age, years, median (IQR)** | 35 (25.3–43) | 36.5 (28–47) |
| **Race/ethnicity, *n* (%)** |  |  |
| White | 15 (19.7) | 27 (14.7) |
| Black | 1 (1.3) | 8 (4.4) |
| Mixed | 48 (63.2) | 139 (75.5) |
| Indigenous | 4 (5.3) | 3 (1.6) |
| Yellow | - | 1 (0.5) |
| Unknown | 8 (10.5) | 5 (2.7) |
| Missing | - | 1 (0.5) |
| **Highest formal educational attendance, *n* (%)** |  |  |
| University/post-graduation | 16 (21.1) | 59 (32.1) |
| High school/technical college | 43 (56.6) | 93 (50.5) |
| Lower secondary | 7 (9.2) | 11 (6.0) |
| Primary | 9 (11.8) | 20 (10.9) |
| Less than primary | 1 (1.3) | 1 (0.5) |
| **Marital Status, *n* (%)** |  |  |
| Single | 32 (42.1) | 86 (46.7) |
| Married | 22 (28.9) | 59 (32.1) |
| Living with a partner | 19 (25.0) | 22 (12.0) |
| Separated or Divorced | 3 (3.9) | 14 (7.6) |
| Widowed | - | 3 (1.6) |
| **Household contacts, median (IQR)** | 3 (2–4) | 2 (1–4) |
| **Self-reported history of chronic diseases, *n* (%)** |  |  |
| Diabetes mellitus | 3 (3.9) | 1 (0.5) |
| Hypertension | 7 (9.2) | 19 (10.3) |
| Joint Disease | 1 (1.3) | 5 (2.7) |
| HIV infection | 1 (1.3) | - |
| Chronic hepatitis C | - | 1 (0.5) |
| Other chronic diseases* | 8 (10.5) | 20 (10.9) |
| **BMI, median (IQR)** | 28.9 (25.2–33.1) | 27 (23.2–29.9) |
| Underweight | - | 3 (1.6) |
| Normal | 17 (22.4) | 49 (26.6) |
| Overweight | 21 (27.6) | 72 (39.1) |
| Class I Obesity | 23 (30.3) | 40 (21.7) |
| Class II Obesity | 7 (9.2) | 13 (7.1) |
| Class III Obesity | 8 (10.5) | 3 (1.6) |
| Missing | - | 4 (2.2) |
| **History of blood transfusion, yes, *n* (%)** | 1(1.3) | 11 (6.0) |
| **History of allergies, yes, *n* (%)** | 16 (21.1) | 40 (21.7) |
| **History of arbovirus infection, yes, *n* (%)** |  |  |
| Zika | - | - |
| Dengue | 5 (6.6) | 28 (15.2) |
| Chikungunya | - | 2 (1.1) |

(*Continued*)

**Table 3.** (Continued)

| Characteristics | Male | Female |
|---|---|---|
| | (*n* = 76) | (*n* = 184) |
| Yellow fever | - | - |

IQR: Interquartile Range, BMI: Body mass index, in kg/m$^2$

*Participants may have more than one chronic disease but were analyzed only once regarding the presence or not of chronic disease. Other chronic diseases reported: sickle cell disease (n = 1), breast cancer (n = 1), depression (n = 2), hypothyroidism (n = 1), hyperthyroidism (n = 1), endometriosis (n = 1), cholelithiasis (n = 1), labyrinthitis (n = 2), gastritis (n = 3), seizures (n = 1), hemorrhoids (n = 1), chronic urticaria (n = 2), contact dermatitis (n = 1), chronic sinusitis (n = 1), rotator cuff syndrome, synovitis, and tenosynovitis (n = 1), multiple myeloma (n = 1), chronic migraine (n = 1), herniated disc disease (n = 2), cardiopathy (n = 3).

**Table 4. Signs and symptoms at enrollment since the onset of disease of 260 study participants of the ZIKABRA study, by sex.**

| Signs/Symptoms | Male n (%) | Female n (%) |
|---|---|---|
| | (*n* = 76) | (*n* = 184) |
| Macular or papular rash* | 75 (98.7) | 183 (99.5) |
| Fever | 72 (94.7) | 163 (88.6) |
| Itching | 72 (94.7) | 182 (98.9) |
| Arthralgia | 66 (86.8) | 176 (95.7) |
| Nonpurulent conjunctivitis | 62 (81.6) | 155 (84.2) |
| Periarticular Edema | 51 (67.1) | 166 (90.2) |
| Headache | 48 (63.2) | 161 (87.5) |
| Myalgia | 43 (58.9) | 107 (60.1) |
| Chills | 39 (52.0) | 116 (63.4) |
| Numbness | 26 (34.2) | 88 (48.1) |
| Nausea | 25 (32.9) | 74 (40.2) |
| Photophobia | 22 (28.9) | 97 (52.7) |
| Lymphonode enlargement | 21 (27.6) | 55 (29.9) |
| Tingling | 18 (23.7) | 56 (30.6) |
| Retro-orbital pain | 17 (22.4) | 74 (40.2) |
| Sweating | 16 (21.1) | 44 (23.9) |
| Prostration | 16 (21.1) | 54 (29.5) |
| Anorexia | 15 (19.7) | 66 (35.9) |
| Diarrhea | 15 (19.7) | 56 (30.4) |
| Oropharyngeal pain | 13 (17.1) | 35 (19.0) |
| Taste alteration | 13 (17.1) | 41 (22.4) |
| Burning | 13 (17.1) | 40 (21.9) |
| Cough | 10 (13.2) | 21 (11.4) |
| Abdominal pain | 8 (10.5) | 37 (20.1) |
| Hoarseness | 7 (9.2) | 16 (8.7) |
| Nasal congestion | 4 (5.3) | 14 (7.6) |
| Dyspnea | 4 (5.3) | 15 (8.2) |
| Coryza | 3 (3.9) | 14 (7.6) |
| Earache | 2 (2.7) | 16 (8.7) |
| Mouth ulcers | 1 (1.4) | 17 (9.6) |
| Dysuria | 1 (1.3) | 8 (4.3) |

(*Continued*)

**Table 4.** (Continued)

| Signs/Symptoms | Male n (%) | Female n (%) |
|---|---|---|
| | (*n* = 76) | (*n* = 184) |
| Bleeding (Epistaxis, Ginvival, metrorrhagia) | - | 6 (3.3) |
| Jaundice | - | 1 (0.6) |
| Vomiting | - | 9 (4.9) |

Male missing: chills (n = 1), earache (n = 1), myalgia (n = 3), mouth ulcers (n = 3). Female missing: chills (n = 1), jaundice (n = 6), numbness (n = 1), burning (n = 1), tingling (n = 1), prostration (n = 1), taste alteration (n = 1), myalgias (n = 6), mouth ulcers (n = 6).

* Two asymptomatic participants were included as contacts (one man and one women).

Full genome sequences will be obtained from several samples which will allow us to study the evolution of the virus within a patient during the disease. Genetic changes over time in consensus sequences and intra-host nucleotide variants can be investigated in different body compartments. Assays for cellular and humoral immune responses will be measured with the blood samples stored during the study. Studies will be also carried out on genetic polymorphisms within host genes reported to influence the rates of pathogen acquisition and/or disease severity.

**Table 5. Physical examination at enrollment of 260 study participants of the ZIKABRA study, by sex.**

| Physical signs | Male n (%) | Female n (%) |
|---|---|---|
| | (*n* = 76) | (*n* = 184) |
| Dehydration | 6 (7.9) | 11 (6.0) |
| Pale skin/mucosa | 1 (1.3) | 14 (7.6) |
| Macular rash | 12 (15.8) | 36 (19.6) |
| Maculopapular rash | 18 (23.7) | 39 (21.2) |
| Skin erythema | 5 (6.6) | 10 (5.4) |
| Vesicular rash | 1 (1.3) | - |
| Petechiae/purpura/Ecchymosis | - | 3 (1.6) |
| Enathema | 1 (1.3) | - |
| Oropharyngeal redness | 1 (1.3) | - |
| Mouth ulcers | - | 1 (0.5) |
| Nonpurulent conjunctivitis | 24 (31.6) | 43 (23.4) |
| Muscular weakness | 3 (3.9) | 8 (4.3) |
| Coarse crackles | 1 (1.3) | - |
| Cardiac murmur | - | 1 (0.5) |
| Abdominal pain | 1 (1.3) | 2 (1.1) |
| Periarticular edema | 9 (11.8) | 39 (21.2) |
| Lymphadenopathy | 21(27.6) | 67 (36.4) |
| Cervical | 8 (38.0) | 33 (49.2) |
| Retroauricular | 8 (38.0) | 32 (47.8) |
| Occipital | 3 (14.3) | 5 (7.5) |
| Submandibular | 1 (4.8) | - |
| Axillar | 2 (9.5) | 3 (4.5) |
| Supraclavicular | - | 1 (1.5) |
| Inguinal | 8 (38.0) | 18 (26.9) |

**Table 6. Characteristics associated with discontinuation in the ZIKABRA cohort study.**

| Characteristics | n | Remained Participants n (%) | Discontinued Participants n (%) | p |
|---|---|---|---|---|
| **Age** | 260 | | | 0.012 |
| Up to 35 | | 101 (45.3) | 25 (67.6) | |
| > 35 | | 122 (54.7) | 12 (32.4) | |
| **Sex** | 260 | | | 0.043 |
| Male | | 60 (26.9) | 16 (43.2) | |
| Female | | 163 (73.1) | 21 (56.8) | |
| **Race/ethnicity** | 246 | | | 0.425 |
| Non-white | | 175 (83.7) | 29 (78.4) | |
| White | | 33 (16.3) | 8 (21.6) | |
| **Formal education** | 260 | | | 0.370 |
| Lower secondary and less | | 44 (19.7) | 5 (13.5) | |
| High school and more | | 179 (80.3) | 32 (86.5) | |
| **Marital Status** | 260 | | | 0.898 |
| Single/separated/divorced/widowed | | 118 (52.9) | 20 (54.1) | |
| Married/living with a partner | | 105 (47.1) | 17 (45.9) | |
| **Underlying chronic medical condition** | 260 | | | 0.313 |
| Yes | | 46 (20.6) | 5 (13.5) | |
| No | | 177 (79.4) | 32 (86.5) | |
| **BMI** | 256 | | | 0.827 |
| Underweight/overweight/obese | | 162 (74.0) | 28 (75.7) | |
| Normal | | 57 (26.0) | 9 (24.3) | |

BMI: Body mass index.

The adherence to the collection of biological samples in the study was substantial, considering the number of visits and sample collection entailed, especially highlighting the collection of rectal swabs, which can be considered a taboo by men in some parts of Brazil.

A biorepository was established in the study centers with a wide range of specimen types creating a unique opportunity to promote future collaborations and data sharing.

So far, 2 manuscripts were published related to this study cohort. The study protocol [12] and an unprecedented publication on detection, persistence, and isolation of ZIKV in rectal swab samples [2].

## Limitations

The study included participants with ZIKV infection in two regions of Brazil to also explore the possible genetic diversity of the virus and influence on viral persistence. However, due to the low circulation of ZIKV in Brazil after the 2016 outbreak, and later circulation of the virus in the northern region of the country, the center in Manaus (Amazonas) virtually included almost all the study participants [15]. As a result, the study will not have the opportunity to study genetic diversity across different regions in the country. Additionally, the final sample size of 282 males and females was not reached, limiting some statistical analysis. This sample size was estimated for evaluation of the primary outcome which is the overall ZIKV persistence rate by 12 months (taking into consideration a design/clustering effect and loss to follow up).

Another limitation of the study that somewhat delayed the start of processing of the study samples was the need to validate some kits and diagnostic tests in fluids that had not been previously validated and somehow intermittent availability of some diagnostic kits and funding provided by the Ministry of Health due to the bureaucratic process.

Finally, the impossibility of following pregnant women in the study and outcomes of ZIKV infection in babies in this cohort. After careful evaluation of risks and benefits of the research for pregnant women, and considering the frequency of collection of body fluids, pregnancy was an exclusion criterion of the study. Following the same principle, women that became pregnant during the study had their participation discontinued. They were, however, promptly referred to existing specialized units for the care of pregnant women with Zika, where adequate care for their special needs and followed up to delivery was provided. All information and exams related to this patient and collected during the study period were shared with the referred unit. Besides, some pregnant patients were invited to participate in ongoing cohort studies of pregnant women infected with ZIKV available in the city where the study was conducted.

## Supporting information

**S1 File. PRE: Pre-screening form.**
(PDF)

**S2 File. TRI: Screening form.**
(PDF)

**S3 File. IDE: Identification form.**
(PDF)

**S4 File. REC: Enrollment form.**
(PDF)

**S5 File. AMO: Sample collection form.**
(PDF)

**S6 File. RES: Body fluids results form.**
(PDF)

**S7 File. RAP: Rapid tests/serology form.**
(PDF)

**S8 File. CLA: Clinical questionnaire a form.**
(PDF)

**S9 File. SEG: Follow-up form.**
(PDF)

**S10 File. CLB: Clinical questionnaire B form.**
(PDF)

**S11 File. FIM: End of the study form.**
(PDF)

## Acknowledgments

The ZIKABRA study team would like to thank all patients who participate in the research. Also, the healthcare workers in Recife who helped us by providing an exclusive/private space at the Hospital das Clínicas for the study to be carried out (Frederico Jorge Ribeiro, Celia Maria Machado Barbosa de Castro, Ana Maria Menezes Caetano, Jose Angelo Rizzo. The healthcare workers at the UPA Caxanga where patients were screened (Audrey Vasconcelos and Georgia Assunção), the Secretary of Health from Rio de Janeiro (Cristina Lemos), and Recife (Jailson Correa) who provided the epidemiological situation of the Zika virus disease.

And finally to the ZIKABRA Study Team (in alphabetical order): Abreu, André Luiz de (General Coordination of Public Health Laboratories (CGLAB/DAEVS/SVS/MS), Brasília-DF, Brazil); Bermudez, Ximena Pamela Diaz (Department of Public Health, University of Brasília, Brasília-DF, Brazil); Bôtto-Menezes, Camila Helena Aguiar (Department of Malaria, Tropical Medicine Foundation Doctor Heitor Vieira Dourado (FMT-HVD), Manaus, Amazonas, Brazil & School of Health Sciences, Amazonas State University (UEA), Manaus, Amazonas, Brazil); Brasil, Patrícia (Acute Febrile Illnesses Laboratory, Evandro Chagas National Institute of Infectious Diseases, Oswaldo Cruz Foundation, Rio de Janeiro, Rio de Janeiro, Brazil); Brito, Carlos Alexandre Antunes (Clinical Hospital of Federal University, Department of Internal Medicine, Recife, Pernambuco, Brazil); Broutet, Nathalie Jeanne Nicole (Department of Sexual and Reproductive Health and Research, World Health Organization, Geneva, Switzerland–**lead author, email:** broutetn@who.int; Calvet, Guilherme Amaral (Acute Febrile Illnesses Laboratory, Evandro Chagas National Institute of Infectious Diseases, Oswaldo Cruz Foundation, Rio de Janeiro, Rio de Janeiro, Brazil); Castilho, Marcia da Costa (Department of Malaria, Tropical Medicine Foundation Doctor Heitor Vieira Dourado (FMT-HVD), Manaus, Amazonas, Brazil); Fernandes, Tatiana Jorge (Acute Febrile Illnesses Laboratory, Evandro Chagas National Institute of Infectious Diseases, Oswaldo Cruz Foundation, Rio de Janeiro, Rio de Janeiro, Brazil); Filippis, Ana Maria Bispo de (Flavivirus Laboratory, Oswaldo Cruz Institute, Oswaldo Cruz Foundation, Rio de Janeiro, Rio de Janeiro, Brazil); Franca, Rafael Freitas Oliveira (Department of Virology and Experimental Therapy, Institute Aggeu Magalhães, Oswaldo Cruz Foundation, Recife, Pernambuco, Brazil); Giozza, Silvana Pereira (Department of Chronic Condition Diseases and Sexually Transmitted Infections, Health Surveillance Secretariat, Ministry of Health, Brazil); Habib, Ndema (Department of Sexual and Reproductive Health and Research, World Health Organization, Geneva, Switzerland); Kara, Edna Oliveira (Department of Sexual and Reproductive Health and Research, World Health Organization, Geneva, Switzerland); Lacerda, Marcus Vinicius Guimarães (Department of Malaria, Tropical Medicine Foundation Doctor Heitor Vieira Dourado (FMT-HVD), Manaus, Amazonas, Brazil & Instituto Leônidas & Maria Deane, Oswaldo Cruz Foundation, Manaus, Amazonas, Brazil); Landoulsi, Sihem (Department of Sexual and Reproductive Health and Research, World Health Organization, Geneva, Switzerland); Lima, Morganna Costa (Department of Virology and Experimental Therapy, Institute Aggeu Magalhães, Oswaldo Cruz Foundation, Recife, Pernambuco, Brazil); Lima, Noemia (Emerging Infectious Diseases Branch, Walter Reed Army Institute of Research, Silver Spring, MD, United States of America); Mello, Maeve Brito de (Regional Advisor for HIV and STI Prevention, Pan American Health Organization (PAHO), WDC, United States of America); Meurant, Robyn (Department of Sexual and Reproductive Health and Research, World Health Organization, Geneva, Switzerland); Modjarrad, Kayvon (Emerging Infectious Diseases Branch, Walter Reed Army Institute of Research, Silver Spring, MD, United States of America); Neto, Armando Menezes (Department of Virology and Experimental Therapy, Institute Aggeu Magalhães, Oswaldo Cruz Foundation, Recife, Pernambuco, Brazil); Pereira, Gerson Fernando (Department of Chronic Condition Diseases and Sexually Transmitted Infections, Health Surveillance Secretariat, Ministry of Health, Brazil), Pimenta, Cristina (Department of Chronic Condition Diseases and Sexually Transmitted Infections, Health Surveillance Secretariat, Ministry of Health, Brazil); Storme, Casey (Emerging Infectious Diseases Branch, Walter Reed Army Institute of Research, Silver Spring, MD, United States of America); Ströher, Ute (Regulation and Prequalification Department, World Health Organization, Geneva, Switzerland); Thorson, Anna (Department of Sexual and Reproductive Health and Research, World Health Organization, Geneva, Switzerland); Trautman, Lydie (US Military HIV Research Program, Walter Reed Army Institute of Research, Silver Spring, Maryland, United States of America).

## Author Contributions

**Conceptualization:** Guilherme Amaral Calvet, Edna Oliveira Kara, Sihem Landoulsi, Ndema Habib, Camila Helena Aguiar Bôtto-Menezes, Rafael Freitas de Oliveira Franca, Marcia da Costa Castilho, Silvana Pereira Giozza, Ximena Pamela Díaz Bermúdez, Kayvon Modjarrad, Patrícia Brasil, Marcus Vinicius Guimarães de Lacerda, Ana Maria Bispo de Filippis, Nathalie Jeanne Nicole Broutet.

**Data curation:** Guilherme Amaral Calvet, Sihem Landoulsi, Ndema Habib, Camila Helena Aguiar Bôtto-Menezes, Rafael Freitas de Oliveira Franca, Armando Menezes Neto, Marcia da Costa Castilho, Tatiana Jorge Fernandes, Ana Maria Bispo de Filippis.

**Formal analysis:** Guilherme Amaral Calvet, Rafael Freitas de Oliveira Franca, Armando Menezes Neto, Marcia da Costa Castilho, Tatiana Jorge Fernandes, Kayvon Modjarrad, Ana Maria Bispo de Filippis.

**Funding acquisition:** Guilherme Amaral Calvet, Edna Oliveira Kara, Gerson Fernando Pereira, Silvana Pereira Giozza, Kayvon Modjarrad, Marcus Vinicius Guimarães de Lacerda, Ana Maria Bispo de Filippis, Nathalie Jeanne Nicole Broutet.

**Investigation:** Guilherme Amaral Calvet, Edna Oliveira Kara, Camila Helena Aguiar Bôtto-Menezes, Rafael Freitas de Oliveira Franca, Armando Menezes Neto, Marcia da Costa Castilho, Kayvon Modjarrad, Noemia Lima, Ana Maria Bispo de Filippis.

**Methodology:** Guilherme Amaral Calvet, Edna Oliveira Kara, Sihem Landoulsi, Ndema Habib, Camila Helena Aguiar Bôtto-Menezes, Rafael Freitas de Oliveira Franca, Armando Menezes Neto, Marcia da Costa Castilho, Silvana Pereira Giozza, Ximena Pamela Díaz Bermúdez, Kayvon Modjarrad, Noemia Lima, Patrícia Brasil, Marcus Vinicius Guimarães de Lacerda, Ana Maria Bispo de Filippis, Nathalie Jeanne Nicole Broutet.

**Project administration:** Guilherme Amaral Calvet, Edna Oliveira Kara, Camila Helena Aguiar Bôtto-Menezes, Rafael Freitas de Oliveira Franca, Marcia da Costa Castilho, Silvana Pereira Giozza, Kayvon Modjarrad, Marcus Vinicius Guimarães de Lacerda, Ana Maria Bispo de Filippis, Nathalie Jeanne Nicole Broutet.

**Resources:** Guilherme Amaral Calvet, Edna Oliveira Kara, Sihem Landoulsi, Ndema Habib, Camila Helena Aguiar Bôtto-Menezes, Rafael Freitas de Oliveira Franca, Marcia da Costa Castilho, Silvana Pereira Giozza, Kayvon Modjarrad, Marcus Vinicius Guimarães de Lacerda, Ana Maria Bispo de Filippis.

**Software:** Guilherme Amaral Calvet, Edna Oliveira Kara, Sihem Landoulsi, Camila Helena Aguiar Bôtto-Menezes, Rafael Freitas de Oliveira Franca, Armando Menezes Neto, Marcia da Costa Castilho, Tatiana Jorge Fernandes, Kayvon Modjarrad, Marcus Vinicius Guimarães de Lacerda, Ana Maria Bispo de Filippis.

**Supervision:** Guilherme Amaral Calvet, Edna Oliveira Kara, Sihem Landoulsi, Camila Helena Aguiar Bôtto-Menezes, Rafael Freitas de Oliveira Franca, Kayvon Modjarrad, Marcus Vinicius Guimarães de Lacerda, Ana Maria Bispo de Filippis, Nathalie Jeanne Nicole Broutet.

**Validation:** Guilherme Amaral Calvet, Edna Oliveira Kara, Rafael Freitas de Oliveira Franca, Armando Menezes Neto, Marcia da Costa Castilho, Kayvon Modjarrad, Ana Maria Bispo de Filippis.

**Visualization:** Guilherme Amaral Calvet, Edna Oliveira Kara, Ana Maria Bispo de Filippis.

**Writing – original draft:** Guilherme Amaral Calvet, Edna Oliveira Kara.

**Writing – review & editing:** Guilherme Amaral Calvet, Edna Oliveira Kara, Sihem Landoulsi, Ndema Habib, Camila Helena Aguiar Bôtto-Menezes, Rafael Freitas de Oliveira Franca, Armando Menezes Neto, Marcia da Costa Castilho, Tatiana Jorge Fernandes, Gerson Fernando Pereira, Silvana Pereira Giozza, Ximena Pamela Díaz Bermúdez, Kayvon Modjarrad, Noemia Lima, Patrícia Brasil, Marcus Vinicius Guimarães de Lacerda, Ana Maria Bispo de Filippis, Nathalie Jeanne Nicole Broutet.

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
