## [Decision Letter · Decision Letter 0]

9 Oct 2020

PONE-D-20-28958

Cohort profile: Study on the persistence of Zika virus in body fluids of patients with ZIKV infection in Brazil (ZIKABRA Study)

PLOS ONE

Dear Dr. Calvet,

Thank you for submitting your manuscript to PLOS ONE. After careful consideration, we feel that it has merit but does not fully meet PLOS ONE’s publication criteria as it currently stands. Therefore, we invite you to submit a revised version of the manuscript that addresses the points raised during the review process.

A large rewritting is requested taking in accoun the general comment from the reviewer 2. As the persistence is the main item analysed here the virology data (ie RNA detection or virus isolation) cannot be partialy skipped as noted by the reviewer 2. The description of the cohort building may be improved by a flowchart as suggested by reviewer 1. Altogether the comments of the 2 reviewers pointed out a paper that deserved to be improved by some clarification in the material and method (and mainly the cohort description) and more carefully linked with the former of future papers that the authors plane to do with this work to improve the discussion conclusion.

We look forward to receiving your revised manuscript.

Kind regards,

Pierre Roques, Ph.D.

Academic Editor

PLOS ONE

Journal Requirements:

3. One of the noted authors is a group or consortium [ZIKABRA Study Team]. In addition to naming the author group, please list the individual authors and affiliations within this group in the acknowledgments section of your manuscript. Please also indicate clearly a lead author for this group along with a contact email address.

Additional Editor Comments (if provided):

Reviewers' comments:

Reviewer's Responses to Questions

**Comments to the Author**

1. Is the manuscript technically sound, and do the data support the conclusions?

Reviewer #1: Yes

Reviewer #2: Partly

2. Has the statistical analysis been performed appropriately and rigorously? 

Reviewer #1: N/A

Reviewer #2: Yes

3. Have the authors made all data underlying the findings in their manuscript fully available?

Reviewer #1: No

Reviewer #2: Yes

4. Is the manuscript presented in an intelligible fashion and written in standard English?

Reviewer #1: No

Reviewer #2: Yes

5. Review Comments to the Author

Reviewer #1: It is great to see a study of this important health issue in Brazil. This is an important arbovirus in Latin America and elsewhere. However, there are some issues that need to be addressed.

Line 89. Screening and recruitment in the cohort: How were the patients recruited? How were the participants referred to the screening locations? Additional details should be provided elaborating upon the information in reference 12.

Line 98. Here it is mentioned that no participants were recruited from Rio de Janeiro. Elsewhere the study alludes to recruitment in three cities. As there was no recruitment from Rio de Janeiro, perhaps Rio de Janeiro should not be mentioned in the methods at all. Furthermore, the statement that there was difference in incidence among the cities should be justified by citing a reference.

The authors should list the eligibility criteria in greater detail, and provide a list of the inclusion and exclusion criteria. For example, the text suggests that the most important symptom was rash. However, in Table 3, not all participants had rash.

ICFs are mentioned on line 107 and line 122. Were there two ICFs? Please clarify. Perhaps a flowchart would make this clearer.

Line 116. “the management was done according to regular procedures”. The authors should explain what is meant by “regular procedures”.

Line 120. “Individuals with rRT-PCR…”. Was the PCR qualitative or quantitative? If the latter, please list the cycle threshold used to classify a sample as positive.

Line 169. The authors repeatedly refer to “Household/sexual contact”. Presumably these two types of contacts are distinct. The study would benefit from clarifying the differences between these two types of contacts.

Line 184. Future collaborations. This section of the text is written in the future tense. The previous sections were written in the present tense. It would strengthen the manuscript to reword this text to use the same tense consistently.

Line 221. Statistical Analysis. The information about the loss to follow-up after three attempts to contact does not fit in well with the data on statistical methods. This could be moved to the section about Methods: Enrollment.

Results

Line 261. Currently, the text states that an individual was classified as having prior exposure to malaria if he or she reported malaria in the previous 30 days. This definition may be too restrictive. The definition of the malaria and Yellow Fever could be improved by stating that patients were assessed for recent malaria or Yellow Fever, without limiting the diagnosis to the last 30 days.

Line 293. Here it is reported that maculopapular rash was the “most important” symptom. It would make the text more precise if the authors were to list the frequency of rash among the participants, which is given in Table 3.

Line 366. Here it is stated that during the recruitment period incidence was low in Brazil. I suggest citing a reference to justify this.

Table 2. What is the difference between “Don’t known”, “missing”, and “other”? I suggest replacing “don’t known” with “unknown”. Moreover, it would strengthen the manuscript to list the sample size N and the number of missing. It might benefit the study to compare demographic characteristics of the positive and negative individuals. E.g. was average age different between ZIKV positive and negative individuals?

Perhaps the study could mention that the persistence of the virus in sperm is important as it could potentially prolong the persistence of ZIKV in the population. For example, during the Ebola epidemic in West Africa in 2016, the virus was spread both by sexual intercourse as well as by other forms of physical contact. Epidemiological studies demonstrated that the first wave of the epidemic was driven by transmission due to physical contact that was not sexual in nature. Subsequent smaller outbreaks were attributed to sexual transmission. There may be parallels with the transmission of Zika by two different routes: mosquito transmission and sexual intercourse.

In connection with this, sexual contacts should be added to the first box of Figure 1.

Reviewer #2: Reviewer report- Cohort profile: Study on the persistence of Zika virus in body fluids of patients with ZIKV infection in Brazil

Summary

Authors of the paper provide a reporting on a cohort that enrolled acute ZIKV patients and their symptomatic and asymptomatic household/sexual contacts. The name of the cohort is ZIKABRA. It is a prospective cohort supported by the WHO and conducted in Brazil to assess the presence and maintenance of ZIKV in human body fluids. Several related markers of infection will be evaluated in the cohort.

Overall impression

Herein, the authors provide an exhaustive reporting of clinical data of participants enrolled in the ZIKABRA cohort. All models of recording questionnaire frames used in the study are available and the tables contains all results on the clinical status and characteristics of the participants.

However, the reporting lacks to developpe specific data and tables about the presence of the virus itself (RNA or virus isolation, as author implement their study especially in places with facilities allowing isolation and detection of the virus). There is only one paragraph, named “Biological specimens”, that shortly summarized the presence of the RNA virus in some fluids.

There isn’t any more information instead of the percentage value of positive participants.

The authors have already published 2 papers on this cohorts and so they will probably publish paper on each fluid separately, but it should be nice to have more discussion on the results reported in this paper.

For example, we haven’t any information of the % of positive male/female for each kind of fluids (obviously apart for semen, breast milk or vaginal secretions). We don’t know neither the positive participants belong to the index case participants or household/sexual contacts, nor the % that correspond to the symptomatic/asymptomatic.

Authors noted some information but failed to develop hypothesize about these facts. For instance, they noted that an important part of overweight people in participants, is there any link with disease presentation or the presence of the virus in fluids?

Authors issued the potential alternative diagnostic tool due to detection of ZIKV RNA in several fluids in the abstract section but they don’t give more information in the main text about this interesting outcome.

Then, authors reported some difficulties to fix the detection of ZIKV RNA, which kind of difficulties to overcome?

I invite authors to go little deeper in the analyze of the results presented in this reporting and link these ones with the presence of the virus in body fluids.

Minor issues

Line 1: The title of the first publication Botto-Menezes et al (doi: 10.3201/eid2505.180904) on this cohort was « Study on the persistence of Zika virus (ZIKV) in body fluids of patients with ZIKV infection in Brazil » and the title of the current submitted paper is : « Cohort profile: Study on the persistence of Zika virus in body fluids of patients with ZIKV infection in Brazil ». Both titles are very closed and it this can be confusing for the reader. May you change the title ?

have to be done in results not in discussion.

Line 67-68: “… coinfection like dengue…”. Please authors write “dengue and chikungunya” but right next words they use HIV, Hepatitis B (HBV)… If authors describe the virus and not the desease, please correct with “dengue virus (DENV)” and “chikungunya virus (CHIKV)”.

Line 85: “… chain reaction (rRT-PCR)…” please homogenized the term for Real Time revers transcription polymerase chain reaction in the text because in line 95 RT-PCR is used.

Line 96: used the short form DENV and CHIKV as authors describe viruses (like ZIKV)

Line 131: used the short form HBV and HCV as authors mention above for all the virus (Line 68).

Line 200: “Specific proposals for collaboration are welcome…”, curious statement in a materiel and methods part.

6. PLOS authors have the option to publish the peer review history of their article (what does this mean?). If published, this will include your full peer review and any attached files.

Reviewer #1: No

Reviewer #2: No

---

## [Author Response · Author response to Decision Letter 0]

23 Nov 2020

REVIEWER COMMENTS 

REVIEWER 1

Line 89

Screening and recruitment in the cohort

• How were the patients recruited? 

• How were the participants referred to the screening locations? 

• Additional details should be provided elaborating upon the information in reference 12

 We added a paragraph to detail how the potential participants were identified and referred to the studies sites: “Briefly, the identification of symptomatic subjects took place in the collaborating clinics: Fiocruz ambulatory (Recife), Tropical Medicine Foundation Doctor Heitor Vieira Dourado (FMT-HVD) ambulatory (Manaus), primary health centers (Family Clinics) and 24-hour emergency units located in the areas of the study intervention. At each of these screening sites, a triage system took place to refer individuals meeting the study inclusion criteria for consideration for enrolment in the study sites.”

Line 98

Here it is mentioned that no participants were recruited from Rio de Janeiro. Elsewhere the study alludes to recruitment in three cities. As there was no recruitment from Rio de Janeiro, perhaps Rio de Janeiro should not be mentioned in the methods at all. Furthermore, the statement that there was difference in incidence among the cities should be justified by citing a reference.

Thank you for your pertinent suggestion. The paragraph was modified and the information regarding Rio de Janeiro was suppressed. 

“Patients with ZIKV infection were screened and enrolled in two study sites in Recife and Manaus cities, representing two regions of Brazil (Northeast and North, respectively). The study sites were selected based on the presence of high population density, high circulation of ZIKV, strong community health network and laboratory facilities able to perform viral culture, ZIKV antigen assays, rRT-PCR, IgM/IgG, neutralizing antibodies test (specific for ZIKV, DENV, and CHIKV) and genetic sequencing. However, by the time of the study onset, the number of ZIKV cases dropped dramatically in Recife. As a result, only five participants were recruited in that site between 21 July 2017 and 13 August 2018.” 

The authors should list the eligibility criteria in greater detail, and provide a list of the inclusion and exclusion criteria. For example, the text suggests that the most important symptom was rash. However, in Table 3, not all participants had rash. A new section on inclusion and exclusion criteria has been added to the manuscript. 

“Index screening inclusion and exclusion criteria

Index case screening inclusion criteria: Men and women ≥ 18 years with acute illness presenting rash; having given consent to blood and urine collection for rRT-PCR (real time revere transcription polymerase chain reaction) testing and to be enrolled if the test result in either sample return positive, including attendance to the study clinic for follow-up visits over 12 months comprising 17 visits, and provide body fluid according to the testing schedule. 

Index case screening exclusion criteria: Individuals under 18 years of age; pregnancy; presenting a condition that would not allow reliable informed consent (e.g. alcohol abuse and substance misuse) or lacking mental capacity to consent participation.

Index case enrolment inclusion criteria: Presenting rRT-PCR test positive for ZIKV in blood and/or urine specimens; having given consent for all body fluid collection, as appropriate, according to the testing schedule.

Index case enrolment exclusion criteria: Presenting rRT-PCR negative test results for ZIKV in blood and urine specimens collected during screening.

Household contacts screening and enrollment criteria

Household contact screening inclusion criteria: individual aged 18 years and above; having given consent for rRT-PCR testing in blood and urine and subsequent enrollment if the tests return a positive result, including coming back to all follow-up visits and provide body fluid according to the testing schedule; no intention to move to a place that would not make possible attendance to the study clinic to continue with the study procedures as per protocol. 

Household contact screening exclusion criteria - As per index case.

Household contact enrolment inclusion and exclusion criteria- As per index case.”

After submission, a second review of the CRFs was performed and two women were found to have a history of rash, and consequently this information is provided in table 3. 

In addition, during this process, further two asymptomatic participants with detectable rRT-PCR were included as contacts. This sentence was added in the text for clarification. 

Also, we revised the table 3 to include symptoms that occurred since the acute phase of the disease and modified the title of the table.

Line 107 and 122

ICFs are mentioned on line 107 and line 122. Were there two ICFs? Please clarify. Perhaps a flowchart would make this clearer. 

Two informed consent forms were used in the study, for the screening and enrollment phases. 

We opted for not adding a flow chart and instead include a paragraph stating the use of these two types of Informed consent.

Line 116. 

“the management was done according to regular procedures”. The authors should explain what is meant by “regular procedures”.

The paragraph was changed to: “The screening test results were delivered to the participant by the study nurse. Individuals whose ZIKV rRT-PCR tests returned negative in blood and urine, were referred to the clinic where they were seen in the first place to receive routine clinical standard of care.”

Line 120. 

“Individuals with rRT-PCR…”. Was the PCR qualitative or quantitative? If the latter, please list the cycle threshold used to classify a sample as positive. 

Thank you for this relevant comment. A new section entitled: “ZIKV detection” has been added to the manuscript containing the following information:

“ZIKV detection was performed by rRT-PCR employing a commercial kit namely ZDC, from the Instituto de Tecnologia em Imunobiológicos Biomanguinhos. The kit was approved by the Agência Nacional de Vigilância Sanitária/ANVISA (registry #80142170032). A test was deemed positive when a sample returned within 38 amplification cycles.” 

Line 169

The authors repeatedly refer to “Household/sexual contact”. Presumably these two types of contacts are distinct. The study would benefit from clarifying the differences between these two types of contacts.

Initially, the study intended to distinguish between household contact and sexual partner for potential ancillary study assessing the contribution of sexual transmission. However, this information is not relevant considering the primary objectives of the cohort study. For this reason, the word "sexual" has been removed from the text.

Line 184. Future collaborations. This section of the text is written in the future tense. The previous sections were written in the present tense. It would strengthen the manuscript to reword this text to use the same tense consistently.

Thank you for your remarks with which we completely agree. As we collected several personally identifiers we decided to delete this section form the manuscript. The following information will be added to the manuscript: 

Data Availability: Data underlying the study cannot be made publicly available due to ethical concerns, as data contain several personally identifiable information. Data are available from Oswaldo Cruz Foundation for researchers who meet the criteria for access to confidential data. Contact information: Institutional Ethics and Research Committee of the Evandro Chagas National Institute of Infectious Diseases, email: cep@ini.fiocruz.br or Guilherme Amaral Calvet; email: guilherme.calvet@ini.fiocruz.br

Line 221. 

Statistical Analysis. The information about the loss to follow-up after three attempts to contact does not fit in well with the data on statistical methods. This could be moved to the section about Methods: Enrollment.

We agree with this pertinent comment. The text has been moved to the Methods section.

Results

Line 261 

Currently, the text states that an individual was classified as having prior exposure to malaria if he or she reported malaria in the previous 30 days. This definition may be too restrictive. The definition of the malaria and Yellow Fever could be improved by stating that patients were assessed for recent malaria or Yellow Fever, without limiting the diagnosis to the last 30 days.

Thank you for pointing this out. We have changed the sentence to make it clearer that we have checked the individual’s history of yellow fever and only for malaria within the 30 days prior to enrollment. “No study participants were diagnosed with yellow fever in the past or malaria within 30 days prior to enrollment”. 

Line 293

Here it is reported that maculopapular rash was the “most important” symptom. It would make the text more precise if the authors were to list the frequency of rash among the participants, which is given in Table 3. Having a rash was one of the criteria adopted for screening potential study participants. The sentence was reworded to “Following rash, which was one of the inclusion criteria for index cases, the most reported symptoms in the enrollment visit since the onset of the disease were fever, itching, arthralgia with or without edema, non-purulent conjunctivitis, headache, and myalgia.”

Line 366 

Here it is stated that during the recruitment period incidence was low in Brazil. I suggest citing a reference to justify this.

Thank you for this very appropriate suggestion. We have included a new reference to support the statement made in the text related to the decrease in incidence rates during the study recruitment phase.

[16] Secretaria de Vigilância em Saúde − Ministério da Saúde. Monitoramento dos casos de dengue, febre de chikungunya e febre pelo vírus Zika até a Semana Epidemiológica 52, 2017. Volume 49, N° 2 - 2018. [cited 2020 November 16]. Available from https://antigo.saude.gov.br/images/pdf/2018/janeiro/23/Boletim-2018-001-Dengue.pdf. 

Table 2. 

What is the difference between “Don’t known”, “missing”, and “other”? I suggest replacing “don’t known” with “unknown”.

We apologize for the typo as we were meant to use “don’t know” rather than “don’t known”. The option “Other” referred to any ethnicity not covered by the list provided under the variable Ethnicity. We thank you for the suggestion to use “unknown” which has been accepted and adopted when patients were unable to define their race/ethnicity. The category “Other” was deleted and the only participant reported in the category was considered as “mixed”. “Missing” refers to information not provided in the CRF. The table 2 has been corrected to reflect the changes aforementioned.

Moreover, it would strengthen the manuscript to list the sample size N and the number of missing. 

Thank you for this recommendation. The information on sample size and missing information have been added to the table.

It might benefit the study to compare demographic characteristics of the positive and negative individuals. E.g. was average age different between ZIKV positive and negative individuals?

This is an important point, thank you. We agree with the recommendation and have included a sentence preceding a new table that contains the comparison between the two groups (individuals with positive and negative results for ZIKV). We found statistically significant differences in age and sex among enrolled and not enrolled participants. A comment on this finding has been added to the manuscript. 

Perhaps the study could mention that the persistence of the virus in sperm is important as it could potentially prolong the persistence of ZIKV in the population. For example, during the Ebola epidemic in West Africa in 2016, the virus was spread both by sexual intercourse as well as by other forms of physical contact. Epidemiological studies demonstrated that the first wave of the epidemic was driven by transmission due to physical contact that was not sexual in nature. Subsequent smaller outbreaks were attributed to sexual transmission. There may be parallels with the transmission of Zika by two different routes: mosquito transmission and sexual intercourse.

In connection with this, sexual contacts should be added to the first box of Figure 1.

We thank the information provided by the reviewer on Ebola transmission which we were aware of. However, this will be part of the Discussion section of the paper where we will be reporting the results for persistence in each of the studied body fluids. The objective of the current manuscript is to describe the profile of the cohort and is intended to be used as a reference.

REVIEWER 2

Herein, the authors provide an exhaustive reporting of clinical data of participants enrolled in the ZIKABRA cohort. All models of recording questionnaire frames used in the study are available and the tables contains all results on the clinical status and characteristics of the participants.

However, the reporting lacks to developpe specific data and tables about the presence of the virus itself (RNA or virus isolation, as author implement their study especially in places with facilities allowing isolation and detection of the virus). There is only one paragraph, named “Biological specimens”, that shortly summarized the presence of the RNA virus in some fluids. There isn’t any more information instead of the percentage value of positive participants. The authors have already published 2 papers on this cohorts and so they will probably publish paper on each fluid separately, but it should be nice to have more discussion on the results reported in this paper.

For example, we haven’t any information of the % of positive male/female for each kind of fluids (obviously apart for semen, breast milk or vaginal secretions). We don’t know neither the positive participants belong to the index case participants or household/sexual contacts, nor the % that correspond to the symptomatic/asymptomatic. As informed above, the aim of this manuscript is to report the cohort profile in detail. The results on Zika persistence in different body fluids is the object of another paper that is being prepared and will be submitted soon.

For this reason and to keep consistency with the objective of the current paper, we decided to delete any information on viral persistence included in the text. 

This was an option to avoid a very lengthy article reporting the results on persistence.

Authors noted some information but failed to develop hypothesize about these facts. For instance, they noted that an important part of overweight people in participants, is there any link with disease presentation or the presence of the virus in fluids? Thank you for the insightful comment. We judge that this would be more appropriate for the manuscript on viral persistence where logistic regression will be conducted to assess possible factors associated with viral persistence.

Authors issued the potential alternative diagnostic tool due to detection of ZIKV RNA in several fluids in the abstract section but they don’t give more information in the main text about this interesting outcome.

We decided to delete this text from the manuscript after receiving your pertinent comment. This was not relevant to what we were intending to report in this paper.

Then, authors reported some difficulties to fix the detection of ZIKV RNA, which kind of difficulties to overcome?

The kits used in the study had been validated for serum, saliva and urine. An extra effort was necessary to standardize the methodology for detection in other fluids. In the case of semen, for instance, the viscosity was an important factor and more diluted samples were used to enable detection.

I invite authors to go little deeper in the analyze of the results presented in this reporting and link these ones with the presence of the virus in body fluids.

We have addressed this point in the answer above related publication of persistence results. Thank you for your suggestion and please refer to our answer for similar points made before.

Line 1: 

The title of the first publication Botto-Menezes et al (doi: 10.3201/eid2505.180904) on this cohort was « Study on the persistence of Zika virus (ZIKV) in body fluids of patients with ZIKV infection in Brazil » and the title of the current submitted paper is : « Cohort profile: Study on the persistence of Zika virus in body fluids of patients with ZIKV infection in Brazil ». Both titles are very closed and it this can be confusing for the reader. May you change the title?

have to be done in results not in discussion.

The actual title for the paper referred by the reviewer is “Zika virus in rectal swab samples.” This has been published as an ancillary study by the ZikaBra study team.

In case the intention was to compare the current manuscript with the protocol publication, we would like to highlight that the title adopted was intentional, considering that we are reporting the profile of this cohort. We only added “cohort profile” to make it clear.

Reference: Calvet GA, Kara EO, Giozza SP, Botto-Menezes CHA, Gaillard P, de Oliveira Franca RF, et al. Study on the persistence of Zika virus (ZIKV) in body fluids of patients with ZIKV infection in Brazil. BMC Infect Dis. 2018;18(1):49. Epub 2018/01/24. doi: 10.1186/s12879-018-2965-4. PubMed PMID: 29357841; PubMed Central PMCID: PMCPMC5778641.

Line 67-68: 

“… coinfection like dengue…”. Please authors write “dengue and chikungunya” but right next words they use HIV, Hepatitis B (HBV)… If authors describe the virus and not the desease, please correct with “dengue virus (DENV)” and “chikungunya virus (CHIKV)”.

Thank you for this relevant recommendation. We agree with the reviewer’s points and have corrected the text as suggested.

Line 85: 

“… chain reaction (rRT-PCR)…” please homogenized the term for Real Time revers transcription polymerase chain reaction in the text because in line 95 RT-PCR is used.

Thank you. The text has been corrected accordingly.

Line 96

Used the short form DENV and CHIKV as authors describe viruses (like ZIKV)

Thank you. The text has been corrected accordingly.

Line 131

Used the short form HBV and HCV as authors mention above for all the virus (Line 68).

We agree with the reviewer and have corrected the text as suggested.

Line 200: 

“Specific proposals for collaboration are welcome…”, curious statement in a materiel and methods part.

Thank you for your comment. As previously mentioned, we decided to delete this section.

Yes. As we collected several personally identifiers and sensitive data from the study participants, we ask the editors to consider the following statement to be written in the manuscript, if accepted for publication: 

Data Availability: Data underlying the study cannot be made publicly available due to ethical concerns, as data contain several personally identifiable information. Data are available from Oswaldo Cruz Foundation for researchers who meet the criteria for access to confidential data. Contact information: Institutional Ethics and Research Committee of the Evandro Chagas National Institute of Infectious Diseases, email: cep@ini.fiocruz.br or Guilherme Amaral Calvet; email: guilherme.calvet@ini.fiocruz.br

 3. One of the noted authors is a group or consortium [ZIKABRA Study Team]. In addition to naming the author group, please list the individual authors and affiliations within this group in the acknowledgments section of your manuscript. Please also indicate clearly a lead author for this group along with a contact email address.

Thank you for this recommendation. We listed the individual authors and affiliations and indicated a lead author for the ZIKABRA Study Team in the acknowledgements section:

“And finally to the ZIKABRA Study Team (in alphabetical order): Abreu, André Luiz de (General Coordination of Public Health Laboratories (CGLAB/DAEVS/SVS/MS), Brasília-DF, Brazil); Bermudez, Ximena Pamela Diaz (Department of Public Health, University of Brasília, Brasília-DF, Brazil); Bôtto-Menezes, Camila Helena Aguiar (Department of Malaria, Tropical Medicine Foundation Doctor Heitor Vieira Dourado (FMT-HVD), Manaus, Amazonas, Brazil & School of Health Sciences, Amazonas State University (UEA), Manaus, Amazonas, Brazil); Brasil, Patrícia (Acute Febrile Illnesses Laboratory, Evandro Chagas National Institute of Infectious Diseases, Oswaldo Cruz Foundation, Rio de Janeiro, Rio de Janeiro, Brazil); Brito, Carlos Alexandre Antunes (Clinical Hospital of Federal University, Department of Internal Medicine, Recife, Pernambuco, Brazil); Broutet, Nathalie Jeanne Nicole (Department of Sexual and Reproductive Health and Research, World Health Organization, Geneva, Switzerland – lead author, email: broutetn@who.int; Calvet, Guilherme Amaral (Acute Febrile Illnesses Laboratory, Evandro Chagas National Institute of Infectious Diseases, Oswaldo Cruz Foundation, Rio de Janeiro, Rio de Janeiro, Brazil); Castilho, Marcia da Costa (Department of Malaria, Tropical Medicine Foundation Doctor Heitor Vieira Dourado (FMT-HVD), Manaus, Amazonas, Brazil); Fernandes, Tatiana Jorge (Acute Febrile Illnesses Laboratory, Evandro Chagas National Institute of Infectious Diseases, Oswaldo Cruz Foundation, Rio de Janeiro, Rio de Janeiro, Brazil); Filippis, Ana Maria Bispo de (Flavivirus Laboratory, Oswaldo Cruz Institute, Oswaldo Cruz Foundation, Rio de Janeiro, Rio de Janeiro,Brazil); Franca, Rafael Freitas Oliveira (Department of Virology and Experimental Therapy, Institute Aggeu Magalhães, Oswaldo Cruz Foundation, Recife, Pernambuco, Brazil); Giozza, Silvana Pereira (Department of Chronic Condition Diseases and Sexually Transmitted Infections, Health Surveillance Secretariat, Ministry of Health, Brazil); Habib, Ndema (Department of Sexual and Reproductive Health and Research, World Health Organization, Geneva, Switzerland); Kara, Edna Oliveira (Department of Sexual and Reproductive Health and Research, World Health Organization, Geneva, Switzerland); Lacerda, Marcus Vinicius Guimarães (Department of Malaria, Tropical Medicine Foundation Doctor Heitor Vieira Dourado (FMT-HVD), Manaus, Amazonas, Brazil & Instituto Leônidas & Maria Deane, Oswaldo Cruz Foundation, Manaus, Amazonas, Brazil); Landoulsi, Sihem (Department of Sexual and Reproductive Health and Research, World Health Organization, Geneva, Switzerland); Lima, Morganna Costa (Department of Virology and Experimental Therapy, Institute Aggeu Magalhães, Oswaldo Cruz Foundation, Recife, Pernambuco, Brazil); Lima, Noemia (Emerging Infectious Diseases Branch, Walter Reed Army Institute of Research, Silver Spring, MD, United States of America); Mello, Maeve Brito de (Regional Advisor for HIV and STI Prevention, Pan American Health Organization (PAHO), WDC, United States of America); Meurant, Robyn (Department of Sexual and Reproductive Health and Research, World Health Organization, Geneva, Switzerland); Modjarrad, Kayvon (Emerging Infectious Diseases Branch, Walter Reed Army Institute of Research, Silver Spring, MD, United States of America); Neto, Armando Menezes (Department of Virology and Experimental Therapy, Institute Aggeu Magalhães, Oswaldo Cruz Foundation, Recife, Pernambuco, Brazil); Pereira, Gerson Fernando (Department of Chronic Condition Diseases and Sexually Transmitted Infections, Health Surveillance Secretariat, Ministry of Health, Brazil), Pimenta, Cristina (Department of Chronic Condition Diseases and Sexually Transmitted Infections, Health Surveillance Secretariat, Ministry of Health, Brazil); Storme, Casey (Emerging Infectious Diseases Branch, Walter Reed Army Institute of Research, Silver Spring, MD, United States of America); Ströher, Ute (Regulation and Prequalification Department, World Health Organization, Geneva, Switzerland); Thorson, Anna (Department of Sexual and Reproductive Health and Research, World Health Organization, Geneva, Switzerland); Trautman, Lydie (US Military HIV Research Program, Walter Reed Army Institute of Research, Silver Spring, Maryland, United States of America).”

---

## [Decision Letter · Decision Letter 1]

16 Dec 2020

PONE-D-20-28958R1

Cohort profile: Study on the persistence of Zika virus in body fluids of patients with ZIKV infection in Brazil (ZIKABRA Study)

PLOS ONE

Dear Dr. Calvet,

Thank you for submitting your manuscript to PLOS ONE. After careful consideration, we feel that it has merit but does not fully meet PLOS ONE’s publication criteria as it currently stands. Therefore, we invite you to submit a revised version of the manuscript that addresses the points raised during the review process.

You have to stated correctly the objective of the article in both the title and the abstract in order it reflect the result and description you provide.

We look forward to receiving your revised manuscript.

Kind regards,

Pierre Roques, Ph.D.

Academic Editor

PLOS ONE

Additional Editor Comments (if provided):

To have any chance of publication the title and abstract must reflect the contain of the article and thus cannot claimed any virological result (viral load in fluid) that you deleted from the final version.

I thus suggest either you provide a true description or you will come back with the data you speach about you would like to provide in a future article.

Reviewers' comments:

Reviewer's Responses to Questions

**Comments to the Author**

1. If the authors have adequately addressed your comments raised in a previous round of review and you feel that this manuscript is now acceptable for publication, you may indicate that here to bypass the “Comments to the Author” section, enter your conflict of interest statement in the “Confidential to Editor” section, and submit your "Accept" recommendation.

Reviewer #1: All comments have been addressed

Reviewer #2: (No Response)

2. Is the manuscript technically sound, and do the data support the conclusions?

Reviewer #1: Yes

Reviewer #2: Partly

3. Has the statistical analysis been performed appropriately and rigorously? 

Reviewer #1: I Don't Know

Reviewer #2: N/A

4. Have the authors made all data underlying the findings in their manuscript fully available?

Reviewer #1: Yes

Reviewer #2: Yes

5. Is the manuscript presented in an intelligible fashion and written in standard English?

Reviewer #1: Yes

Reviewer #2: Yes

6. Review Comments to the Author

Reviewer #1: (No Response)

Reviewer #2: Authors answer or/and corrected minor suggestions.

However, in the answer to comments, they mentioned that the purpose/objective of this paper was only to report the profile of a cohort and delete any information on viral persistence, submitting results and virologic data to a future publication dedicated to this purpose. They answered only to the minor comments, stayed evasive and avoided the question relative to presence of ZIKV keeping this point for a future publication.

I therefore note that this paper is only about the cohort profile but not on ZIKV persistence in body fluid contrary to what the title suggests.

7. PLOS authors have the option to publish the peer review history of their article (what does this mean?). If published, this will include your full peer review and any attached files.

Reviewer #1: No

Reviewer #2: No

---

## [Author Response · Author response to Decision Letter 1]

18 Dec 2020

Authors answer or/and corrected minor suggestions. However, in the answer to comments, they mentioned that the purpose/objective of this paper was only to report the profile of a cohort and delete any information on viral persistence, submitting results and virologic data to a future publication dedicated to this purpose. They answered only to the minor comments, stayed evasive and avoided the question relative to presence of ZIKV keeping this point for a future publication.

I therefore note that this paper is only about the cohort profile but not on ZIKV persistence in body fluid contrary to what the title suggests. 

We agree with this pertinent comment.

We changed the manuscript title to: “Cohort profile: Study on Zika virus infection in Brazil (ZIKABRA Study)”

We also changed the short title and add a sentence in the abstract section to better reflect the objective of the manuscript.

---

## [Editor Report · Decision Letter 2]

21 Dec 2020

Cohort profile: Study on Zika virus infection in Brazil (ZIKABRA Study)

PONE-D-20-28958R2

Dear Dr. Calvet,

We’re pleased to inform you that your manuscript has been judged scientifically suitable for publication and will be formally accepted for publication once it meets all outstanding technical requirements.

Kind regards,

Pierre Roques, Ph.D.

Academic Editor

PLOS ONE
---

## [Editor Report · Acceptance letter]

23 Dec 2020

PONE-D-20-28958R2 

Cohort profile: Study on Zika virus infection in Brazil (ZIKABRA Study) 

Dear Dr. Calvet:

I'm pleased to inform you that your manuscript has been deemed suitable for publication in PLOS ONE. Congratulations! Your manuscript is now with our production department. 

Kind regards, 

on behalf of

Dr. Pierre Roques 

Academic Editor

PLOS ONE